# Adaptive Sampling for Minimax Fair Classification

**Shubhanshu Shekhar** *     **Greg Fields** *     **Mohammad Ghavamzadeh** †     **Tara Javidi** *

## Abstract

Machine learning models trained on uncurated datasets can often end up adversely affecting inputs belonging to underrepresented groups. To address this issue, we consider the problem of adaptively constructing training sets which allow us to learn classifiers that are fair in a *minimax* sense. We first propose an adaptive sampling algorithm based on the principle of *optimism*, and derive theoretical bounds on its performance. We also propose heuristic extensions of this algorithm suitable for application to large scale, practical problems. Next, by deriving algorithm independent lower-bounds for a specific class of problems, we show that the performance achieved by our adaptive scheme cannot be improved in general. We then validate the benefits of adaptively constructing training sets via experiments on synthetic tasks with logistic regression classifiers, as well as on several real-world tasks using convolutional neural networks (CNNs).

## 1   Introduction

Machine learning (ML) models are increasingly being applied for automating the decision-making process in several sensitive applications, such as loan approval and employment screening. However, recent work has demonstrated that discriminatory behaviour might get encoded in the model at various stages of the ML pipeline, such as data collection, labelling, feature selection, and training, and as a result, adversely impact members of some protected groups in rather subtle ways (Barocas and Selbst, 2016). This is why ML researchers have started to introduce a large number of *fairness measures* to include the notion of fairness in the design of their algorithms. Some of the important measures of fairness include demographic parity (Zemel et al., 2013), equal odds and opportunity (Hardt et al., 2016; Woodworth et al., 2017), individual fairness (Dwork et al., 2012), and minimax fairness (Feldman et al., 2015). The *minimax* fairness is particularly important in scenarios in which it is necessary to be as close as possible to equality without introducing unnecessary harm (Ustun et al., 2019). These scenarios are common in areas such as healthcare and predicting domestic violence. A measure that has been explored to achieve this goal is predictive risk disparity (Feldman et al., 2015; Chen et al., 2018; Ustun et al., 2019). Instead of using the common approach of putting constraints on the norm of discrimination gaps, Martinez et al. (2020) has recently introduced the notion of *minimax Pareto fairness*. These are minimax classifiers that are on the Pareto frontier of prediction risk, i.e., no decrease in the predictive risk of one group is possible without increasing the risk of another one.

In this paper, we are primarily interested in the notion of minimax fairness in terms of the predictive risk. However, instead of studying the training phase of the ML pipeline, our focus is on the data-collection stage, motivated by Jo and Gebru (2019) and Holstein et al. (2018). In particular, we study the following question: *given a finite sampling budget, how should a learner construct a training set consisting of elements from different protected groups in appropriate proportions to ensure that a classifier trained on this dataset achieves minimax fairness?*

Our work is motivated by the following scenario: suppose we have to learn a ML model for performing a task (e.g., loan approval) for inputs belonging to different groups based on protected attributes, such

---

*Electrical and Computer Engineering Department at UCSD ({shshekha,grfields,tjavidi}@ucsd.edu)
†Google Research (ghavamza@google.com)

35th Conference on Neural Information Processing Systems (NeurIPS 2021).

as race or gender. Depending upon the distribution of input-label pairs from different groups, the given task may be statistically harder for some groups. Our goal is to ensure that the eventual ML model has optimal predictive accuracy for the worst-off group. We show that, under certain technical conditions, this results in a model with comparable predictive accuracy over all groups. One approach for this would be to train separate classifiers for each group. However, this is often not possible as the group-membership information may not be available at the deployment time, or it may be forbidden by law to explicitly use the protected characteristics as an input in the prediction process (Lipton et al., 2017, § 1). To ensure having a higher proportion of samples from the *harder* groups without knowing the identity of the hard or easy groups apriori, we consider this problem in an *active setting*, where a learner has to incrementally construct a training set by drawing samples one at a time (or a batch at a time) from different groups. Towards this goal, we propose and analyze an adaptive sampling scheme based on the principle of *optimism*, used in bandits literature (e.g., Auer et al. 2002), that detects the *harder* groups and populates the training set with more samples from them in an adaptive manner. We also wish to note that bias in ML has multi-faceted origins and that our work here addresses dataset construction and cannot account for bias introduced by model selection, the underlying data distribution, or other sources as discussed in Hooker (2021), Suresh and Guttag (2019). We also endeavor to ensure *minimax* fairness, but in some contexts another notion of fairness, such as those mentioned above, may be more appropriate or equitable. In general, application of our algorithm is not a guarantee that the resulting model is wholly without bias.

Our *main contributions* are: **1)** We first propose an optimistic adaptive sampling strategy, $\mathcal{A}_{\text{opt}}$, for training set construction in Section 3. This strategy is suited to smaller problems and admits theoretical guarantees. We then introduce a heuristic variant of $\mathcal{A}_{\text{opt}}$ in Section 4 that is more suitable to practical problems involving CNNs. **2)** We obtain upper bounds on the convergence rate of $\mathcal{A}_{\text{opt}}$, and show its minimax near-optimality by constructing a matching lower bound in Section 3.1. **3)** Finally, we demonstrate the benefits of our algorithm with empirical results on several synthetic and real-world datasets in Section 5.

**Related Work.** The closest work to ours is by Abernethy et al. (2020), where they propose an $\epsilon$-greedy adaptive sampling strategy. They present theoretical analysis under somewhat restrictive assumptions and also empirically demonstrate the benefits of their strategy over some baselines. We describe their results in more detail in Appendix C.1, and employ the tools we develop to analyze our algorithm to perform a thorough analysis of the excess risk of their strategy under a much less restrictive set of assumptions and to show some necessary conditions on the value of their exploration parameter, $\epsilon$. We find comparable empirical results for both algorithms given sufficient tuning of their respective hyperparameters, we report some of these results in Section 5 and compare the algorithms in Appendix C. In another related work, Anahideh et al. (2020) propose a fair adaptive sampling strategy that selects points based on a linear combination of model accuracy and fairness measures, and empirically study its performance. These results, however, do not obtain convergence rates of the excess risk of their respective methods and only offer implementations for small-scale datasets.

The above results study the problem of fair classification in an *active* setting and target the *data-collection* stage of the ML pipeline. There are also works that take a *passive* approach to this problem and focus of the *training* phase of the pipeline. Agarwal et al. (2018) design a scheme for learning fair binary classifiers with fairness metrics that can be written as linear constraints involving certain conditional moments. This class of fairness metrics, however, do not contain the minimax fairness measure. Diana et al. (2021) propose a method for constructing (randomized) classifiers for minimax fairness w.r.t. the empirical loss calculated on the given training set. Similarly, Martinez et al. (2020) derive an algorithm for learning a Pareto optimal minimax fair classifier, under the assumption that the learner has access to the true expected loss functions. Thus, the theoretical guarantees in Diana et al. (2021) and Martinez et al. (2020) hold under the assumption of large training sets. Our work, in contrast, constructs the training set incrementally (active setting) from scratch while carefully taking into account the effects of the finite sample size. Finally, we note that the data-collection strategies proposed in our paper can, in principle, be combined with the training methods presented in Martinez et al. (2020) and Diana et al. (2021) to further guarantee the (minimax) fairness of the resulting classifier. We leave the investigation of this approach for future work.

Besides data-collection and training, there have been studies in the fair ML literature on other aspects of the ML pipeline, such as pre-processing (Celis et al., 2020), learning feature representations (Zemel et al., 2013), post-processing (Hardt et al., 2016), and model documentation (Mitchell et al., 2018). We refer readers to a recent survey by Caton and Haas (2020) for detailed description of these results.

## 2 Problem Formulation

Consider a classification problem with the feature (input) space $\mathcal{X}$, label set $\mathcal{Y}$, and a protected attribute set $\mathcal{Z} = \{z_1, \ldots, z_m\}$. For any $z \in \mathcal{Z}$, we use $P_z(x, y)$ as a shorthand for $P_{XY|Z=z}(X = x, Y = y \mid Z = z)$ to denote the feature-label joint distribution given that the protected feature value is $Z = z$. We also use $\mathbb{E}_z[\cdot]$ as a shorthand for the expectation w.r.t. $P_z$. A (non-randomized) classifier is a mapping $f : \mathcal{X} \mapsto \mathcal{Y}$, which assigns a label $y \in \mathcal{Y}$ to every feature $x \in \mathcal{X}$. For a family of classifiers $\mathcal{F} \subset \mathcal{Y}^{\mathcal{X}}$, a loss function $\ell : \mathcal{F} \times \mathcal{X} \times \mathcal{Y} \mapsto \mathbb{R}$, and a mixture distribution over the protected attributes $\pi \in \Delta_m$, we define the $\pi$-*optimal classifier* $f_\pi$ as

$$f_\pi \in \arg\min_{f \in \mathcal{F}} \; \mathbb{E}_\pi\left[\ell(f, X, Y)\right] := \sum_{z \in \mathcal{Z}} \pi(z) \, \mathbb{E}_z\left[\ell(f, X, Y)\right]. \tag{1}$$

When $\pi$ lies on the corners of the simplex $\Delta_m$, i.e., $\pi(z) = 1$ for a $z \in \mathcal{Z}$, we use the notation $f_z$ instead of $f_\pi$. For $\pi$ in the interior of $\Delta_m$, it is clear that the risk of $f_\pi$ on group $z$, i.e., $\mathbb{E}_z\left[\ell(f_\pi, X, Y)\right]$, must be larger than the best possible classification loss for $P_{XY|Z=z}$. In this paper, our goal is to develop an *active sampling scheme* to find the fair mixture distribution $\pi^*$ in a *minimax* sense (*minimizing the maximum risk among the groups*), i.e.,

$$\pi^* \in \arg\min_{\pi \in \Delta_m} \; \max_{z \in \mathcal{Z}} \; L(z, f_\pi) := \mathbb{E}_z\left[\ell(f_\pi, X, Y)\right]. \tag{2}$$

The active sampling problem that we study in this paper can be formally defined as follows:

**Problem 1.** *Suppose $\mathcal{F}$ denotes a family of classifiers, $\ell$ a loss function, $n$ is a sampling budget, and $\mathcal{O} : \mathcal{Z} \mapsto \mathcal{X} \times \mathcal{Y}$ an oracle that maps any attribute $z \in \mathcal{Z}$ to a feature-label pair $(X, Y) \sim P_{XY|Z=z}$. The learner designs an adaptive sampling scheme $\mathcal{A}$ that comprises a sequence of mappings $(A_t)_{t=1}^{n}$ with $A_t : (\mathcal{X} \times \mathcal{Y})^{t-1} \mapsto \mathcal{Z}$ to adaptively query $\mathcal{O}$ and construct a dataset of size $n$. Let $\pi_n \in \Delta_m$ denote the resulting empirical mixture distribution over $\mathcal{Z}$, where $\pi_n(z) = N_{z,n}/n$ and $N_{z,n}$ is the number of times that $\mathcal{A}$ samples from $P_z$ in $n$ rounds. Then, the quality of the resulting dataset is measured by the excess risk, or sub-optimality, of the $\pi_n$-optimal classifier $f_{\pi_n}$, i.e.,*

$$\mathcal{R}_n(\mathcal{A}) := \max_{z \in \mathcal{Z}} \; L\left(z, f_{\pi_n}\right) \; - \; \max_{z \in \mathcal{Z}} \; L\left(z, f_{\pi^*}\right), \tag{3}$$

*where $\pi^*$ is the fair mixture distribution defined by* (2)*. Hence, the goal of the learner is to design a strategy $\mathcal{A}$ which has a small excess risk $\mathcal{R}_n(\mathcal{A})$.*

Informally, the algorithm should adaptively identify the harder groups $z$ and dedicate a larger portion of the overall budget $n$ to sample from their distributions (see Section 2.2 for an illustrative example).

### 2.1 Properties of the Fair Mixture

As discussed above, our goal is to derive an active sampling scheme that allocates the overall budget $n$ over the attributes $z \in \mathcal{Z}$ in a similar manner as the *unknown* fair mixture $\pi^*$, defined by (2). Thus, it is important to better understand the properties of $\pi^*$ and the $\pi$-optimal classifiers $f_\pi$, defined by (1). We state three properties of $f_\pi$ and $\pi^*$ in this section. We refer the readers to Martinez et al. (2020) for the definitions of Pareto front and convexity discussed in this section.

**Property 1.** As discussed in Martinez et al. (2020, Section 4), any $f_\pi$ that solves (1) for a $\pi$ with $\pi(z) > 0$, $\forall z \in \mathcal{Z}$, belongs to the Pareto front $\mathcal{P}_{\mathcal{Z}, \mathcal{F}}$ of the risk functions $\{L(z, f)\}_{z \in \mathcal{Z}}$, $\forall f \in \mathcal{F}$.

**Property 2.** We prove in Proposition 1 (see Appendix A for the proof) that under the following two assumptions on the conditional distributions $\{P_z\}_{z \in \mathcal{Z}}$, function class $\mathcal{F}$, and loss function $\ell$, there exists a unique fair mixture $\pi^*$, whose classifier $f_{\pi^*}$ has equal risk over all attributes $z \in \mathcal{Z}$.

**Assumption 1.** *The mapping $\pi \mapsto L(z, f_\pi)$ is continuous for all attributes $z \in \mathcal{Z}$. Furthermore, if $\pi, \nu \in \Delta_m$ are such that $\pi(z) > \nu(z)$ for an attribute $z \in \mathcal{Z}$, then $L(z, f_\pi) < L(z, f_\nu)$.*

The above assumption indicates that increasing the weight of an attribute $z$ in the mixture distribution must lead to an increase in the performance of the resulting classifier on the distribution $P_z$.

**Assumption 2.** *For any two distinct attributes $z, z' \in \mathcal{Z}$, we must have $L(z, f_z^*) < L(z', f_z^*)$, for any $f_z^* \in \arg\min_{f \in \mathcal{F}} L(z, f)$.*

This assumption requires that any optimal classifier corresponding to distribution $P_z$ to have higher risk w.r.t. the distribution $P_{z'}$ of any other attribute $z' \in \mathcal{Z}$. Note that Assumption 2 may not always hold, for example, if one attribute is *significantly* easier to classify than another.

**Proposition 1.** *Let Assumptions 1 and 2 hold for the conditional distributions $\{P_z\}_{z\in\mathcal{Z}}$, the loss function $\ell$, and the function class $\mathcal{F}$. Then, there exists a unique $\pi^* \in \Delta_m$ that achieves the optimal value for problem* (2) *and satisfies* $L(z_1, f_{\pi^*}) = L(z_2, f_{\pi^*}) = \cdots = L(z_m, f_{\pi^*}) := M^*$.

In other words, Assumptions 1 and 2 define a regime where there exists a fair mixture $\pi^*$, whose classifier $f_{\pi^*}$ achieves complete parity in the classification performance across all attributes. Moreover, Property 1 indicates that $f_{\pi^*}$ also belongs to the Pareto front $\mathcal{P}_{\mathcal{Z},\mathcal{F}}$. Therefore, under these two assumptions, the equal risk classifier not only belongs to the Pareto front, but it is also the minimax Pareto fair classifier (Lemma 3.1 in Martinez et al. 2020). Note that, in general, without Assumption 1, the classifier that attains equality of risk might have worse performance on all attributes than the minimax Pareto fair classifier (Lemma 3.2 in Martinez et al. 2020).

**Property 3.** We may show that when both the function class $\mathcal{F}$ and risk functions $\{L(z,\cdot)\}_{z\in\mathcal{Z}}$ are *convex*, $f_{\pi^*}$ is a minimax Pareto fair classifier. As originally derived by Geoffrion (1968) and then restated by Martinez et al. (2020, Theorem 4.1), under these convexity assumptions, the Pareto front $\mathcal{P}_{\mathcal{Z},\mathcal{F}}$ is convex and any classifier on $\mathcal{P}_{\mathcal{Z},\mathcal{F}}$ is a solution to (1). This together with Property 1 indicates that the classifier $f_{\pi^*}$ corresponding to the fair (minimax optimal) mixture $\pi^*$ is on the Pareto front, and additionally, is a minimax Pareto fair classifier.

## 2.2 Synthetic Models

It is illustrative to consider a class of synthetic models which we use as a running example.

**Definition 1** (SyntheticModel1). Set $\mathcal{X} = \mathbb{R}^2$, $\mathcal{Y} = \{0,1\}$, and $\mathcal{Z} = \{u, v\}$. Each instance of our synthetic model is defined by the set of distributions $P_{XY|Z}$ that satisfy the following: $P_{Y|Z}(Y = 1|Z = z) = 0.5$ for both $z \in \mathcal{Z}$, and $P_{X|Y=y,Z=z} = \mathcal{N}(\mu_{yz}, I_2)$ for all $(y, z) \in \mathcal{Y} \times \mathcal{Z}$. Thus, each instance of this model can be represented by a mean vector $\boldsymbol{\mu} = (\mu_{yz} : y \in \mathcal{Y}, \ z \in \mathcal{Z})$.

The idea behind this class of models is that the 'hardness' of the classification problem for a value of $z \in \mathcal{Z}$ depends on the distance between $\mu_{0z}$ and $\mu_{1z}$. The more separated these two mean vectors are, the easier it is to distinguish the labels. Thus, depending on this distance, it is expected that different protected attributes may require different fractions of samples in the training set to achieve comparable test accuracy. To illustrate this, we consider two instances of the SyntheticModel1: **I.** $\mu_{0u} = (-2, 2)$, $\mu_{1u} = (2, -2)$, $\mu_{0v} = (-1, -1)$, and $\mu_{1v} = (1, 1)$, and **II.** $\mu_{0u} = (-1.5, 1.5)$, $\mu_{1u} = (1.5, -1.5)$, $\mu_{0v} = (-2, -2)$, and $\mu_{1v} = (2, 2)$.

For both model instances, we trained a logistic regression classifier over training sets with 1001 equally spaced values of $\pi(u)$ in $[0, 1]$. Figure 1 shows the test accuracy of the learned classifier for both attributes (blue and red curves) as well as their minimum (black curve) for different values of $\pi(u)$. Since the pair $(\mu_{0u}, \mu_{1u})$ is better separated than $(\mu_{0v}, \mu_{1v})$ in the first instance, it is easier to classify inputs with $Z = u$ than those with $Z = v$, and thus, it requires fewer training samples from $P_{z=u}$ than $P_{z=v}$ to achieve the same accuracy. This is reflected in Figure 1 *(left)* that shows the best (min-max) performance is achieved at $\pi(u) \approx 0.23$. An opposite trend is observed in the second instance (*right* plot in Figure 1), where the mean vectors corresponding to the attribute $Z = v$ are better separated, and hence, require fewer training samples to achieve the same accuracy.

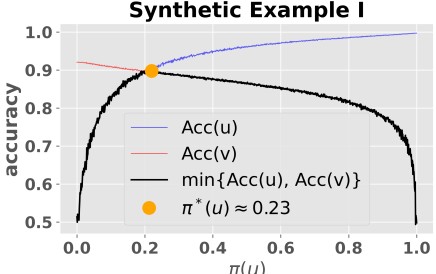
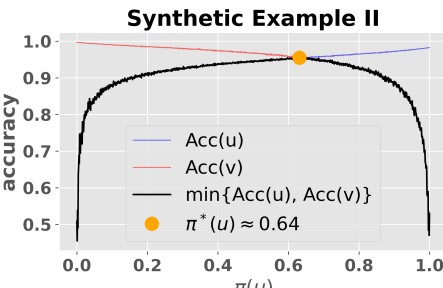

Figure 1: The variation of the minimum prediction accuracy over the two attributes $\mathcal{Z} = \{u, v\}$ with $\pi(u)$ for Instance I, on the left, and Instance II, on the right, of the SyntheticModel1.

# 3 Optimistic Adaptive Sampling Strategy

In this section, we present our *optimistic adaptive sampling* algorithm $\mathcal{A}_{\text{opt}}$ for Problem 1. Algorithm 1 contains a simple pseudo-code for $\mathcal{A}_{\text{opt}}$, see Algorithm 2 for a more detailed, formal pseudo-code. The algorithm proceeds in the following phases:

**Phase 1.** In the initial phase $t \le m = |\mathcal{Z}|$, we draw two independent samples from $P_z = P_{XY|Z=z}$, for each attribute $z \in \mathcal{Z}$, and add one to the training dataset $\mathcal{D}_t$ and the other one to $\mathcal{D}_z$. We use $\mathcal{D}_t$ to learn a common (over all $z \in \mathcal{Z}$) classifier $\hat{f}_t$ via *empirical risk minimization* (ERM). The independent datasets $\{\mathcal{D}_{z_i}\}_{i=1}^m$ are used for estimating the performance of $\hat{f}_t$ for each $z \in \mathcal{Z}$.

**Phase 2.** In each round $t > m$, we choose an attribute $z_t$ according to the following **selection rule**: *if* there exists an attribute $z$ whose number of samples in $\mathcal{D}_t$, denoted by $N_{z_t,t}$, is fewer than $t^\xi$ for some input $\xi \in (0,1)$ (Line 5), we set it as $z_t$ (Line 6), *else* we set $z_t$ as the attribute which has the largest upper confidence bound (UCB) (described below) for the risk of the classifier $f_{\pi_t}$ (Line 8). Here $\pi_t$ denote the empirical mixture distribution (over the attributes $z$) of $\mathcal{D}_t$.

**Phase 3.** When the attribute $z_t$ is selected in *Phase 2*, the algorithm draws a pair of independent samples from $P_{z_t}$, adds one to $\mathcal{D}_t$ and the other to $\mathcal{D}_{z_t}$, and updates $N_{z_t,t}$, $\pi_t$, and the uniform deviation bound $e_{z_t}(N_{z_t,t})$ (described below) (Lines 10 to 12). The updated dataset $\mathcal{D}_t$ is then used to learn a new candidate classifier $\hat{f}_t$ (Line 13). *Phases 2 and 3* are repeated until the sampling budget is exhausted, i.e., $t = n/2$ (note that we sample twice at each round).

**Calculating UCB.** To construct the UCB, we introduce two additional assumptions:

For stating the next assumption, we will use the notations $L(z, f) = \mathbb{E}_z[\ell(f, X, Y)]$ and $L(\pi, f) = \sum_{z \in \mathcal{Z}} \pi(z) L(z, f)$ for any $f \in \mathcal{F}$, $z \in \mathcal{Z}$ and $\pi \in \Delta_m$.

---

**Algorithm 1:** Optimistic Adaptive Sampling for Minimax Fair Classification.

**Input:** $n$ (budget), $\mathcal{F}$ (function class), $\ell$ (loss function), $\xi$ (forced exploration term)

1 **Initialize:** $\mathcal{D}_1 = \emptyset$; $\{\mathcal{D}_{z_i}\}_{i=1}^m = \emptyset$; $\{N_{z_i,1}\}_{i=1}^m = 0$;
2 **for** $t = 1, \ldots, n/2$ **do**
3     **if** $t \le m$ **then**
4         $z_t = z_t$;
5     **else if** $\min_{z \in \mathcal{Z}} N_{z,t} < t^\xi$ **then**
6         $z_t \in \arg\min_{z \in \mathcal{Z}} N_{z,t}$;
7     **else**
8         $z_t = \arg\max_{z \in \mathcal{Z}} U_t(z, \hat{f}_t)$;
        // see Equation 4
9     **end**
10     Draw two independent samples from $P_{z_t}$;
11     Add the first one to $\mathcal{D}_t$ and the second one to $\mathcal{D}_{z_t}$;
12     Update $N_{z_t,t}$; $\pi_t$; $e_{z_t}(N_{z_t,t})$;
13     Learn classifier $\hat{f}_t$ by ERM using $\mathcal{D}_t$;
14 **end**

**Output:** $\pi_{n/2}$ and $\hat{f}_{n/2}$

---

**Assumption 3.** *There exist positive constants $\epsilon_0, C > 0$ such that for any function $f \in \mathcal{F}$ and any $\pi \in \Delta_m$, with $\pi(z) > 0$, $\forall z \in \mathcal{Z}$, if we have $L(\pi, f) \le L(\pi, f_\pi) + \epsilon$, for some $0 < \epsilon \le \epsilon_0$, then $\pi(z) \times |L(z, f) - L(z, f_\pi)| \le 2C\epsilon$, $\forall z \in \mathcal{Z}$.*

Assumption 3 is stated in abstract terms, so Example 1 describes a problem for which it holds.

**Example 1.** *Consider the binary classification problem, with $\mathcal{X} = [0,1]^k$ and $\mathcal{Y} = \{0,1\}$ and $\mathcal{Z} = \{z_1, \ldots, z_m\}$. Let $\mathcal{F}$ be the given family of classifiers $f : \mathcal{X} \to \mathcal{Y}$, and introduce $d(f_1, f_2) = \mu(\{f_1 \ne f_2\})$, where $\mu$ is the Lebesgue measure on $\mathcal{X}$. Assume that the marginal distributions $P_{X|Z=z}$ for all $z \in \mathcal{Z}$ admit densities $\nu_z$ with respect to $\mu$ satisfying $C_1 \le \nu_z(x) \le C_2$ for all $x \in \mathcal{X}$. Note that this is a version of the* strong-density *assumption used in prior works in classification, such as (Audibert and Tsybakov, 2007, Definition 2.2). Then, for the $\ell_{01}$ loss, we observe that the expected loss between any two $f_1, f_2$ satisfies $C_1 d(f_1, f_2) \le |L(z, f_1) - L(z, f_2)| \le C_2 d(f_1, f_2)$.*

*As a result of this, for any $\pi \in \Delta_m$ and $f \in \mathcal{F}$, we have $C_1 d(f, f_\pi) \le L(\pi, f) - L(\pi, f_\pi) \le C_2 d(f, f_\pi)$. Therefore, if it is known that $L(\pi, f) - L(\pi, f_\pi) \le \epsilon$, then we can conclude that we must have $d(f, f_\pi) \le \epsilon/C_1$. Hence, by the* strong-density *assumption on $P_{X|Z=z}$ we have that $\pi(z) \times |L(z, f) - L(z, f_\pi)| \le 1 \times C_2 d(f, f_\pi) \le (C_2/C_1)\epsilon := 2C\epsilon$.*

Our final assumption is a standard uniform convergence requirement which is satisfied by many commonly used families of classifiers (several examples are detailed in Remark 1).

**Assumption 4.** *Let $\delta \in [0,1]$ be a confidence parameter. For each $z \in \mathcal{Z}$, there exists a monotonically non-increasing sequence, $\{e_z(N, \mathcal{F}, \delta) : N \ge 1\}$, with $\lim_{N \to \infty} e_z(N, \mathcal{F}, \delta) = 0$, such that the following event holds with probability at least $1 - \delta/2$:*

$$\Omega = \bigcap_{z \in \mathcal{Z}} \bigcap_{N=1}^{\infty} \left\{ \sup_{f \in \mathcal{F}} \left| \widehat{L}_N(z, f) - L(z, f) \right| \leq e_z(N, \mathcal{F}, \delta) \right\},$$

*where* $\widehat{L}_N(z, f) := \frac{1}{N} \sum_{i=1}^{N} \ell\big(f, X_z^{(i)}, Y_z^{(i)}\big), \ \forall z \in \mathcal{Z}$, *with* $\big(X_z^{(i)}, Y_z^{(i)}\big)_{i=1}^{N}$ *being an i.i.d. sequence of input-label pairs from* $P_z$.

In what follows, we will drop the $\mathcal{F}$ and $\delta$ dependence and refer to $e_z(N, \mathcal{F}, \delta)$ as $e_z(N)$ for all $z \in \mathcal{Z}$ and $N \geq 1$. Given an appropriate sequence of $e_z(N)$, we construct the UCB for the risk function $L(z, f_{\pi_t})$, defined by Equation (2), with the following expression:

$$U_t(z, \hat{f}) := \frac{1}{|\mathcal{D}_z|} \sum_{(x,y) \in \mathcal{D}_z} \ell(\hat{f}_t, x, y) \ + \ e_z(N_{z,t}) + \frac{2C}{\pi_t(z)} \sum_{z' \in \mathcal{Z}} \pi_t(z') e_{z'}(N_{z',t}) . \tag{4}$$

Where $C$ is the constant described in Assumption 3. The first term on the RHS is the empirical loss of the learned classifier at round $t$ on attribute $Z = z$, so by Assumption 4 the first two terms of the RHS of Equation 4 give a high probability upper bound on $L(z, \hat{f}_t)$, the expected loss of $\hat{f}_t$ conditioned on $Z = z$. The third term then provides an upper bound on the difference $|L(z, f_{\pi_t}) - L(z, \hat{f}_t)|$ and so altogether this provides the desired UCB on the expected loss of the $\pi_t$ optimal classifier on attribute $Z = z$. The form of the third term is due to Assumption 3 and is discussed in detail in Appendix B.1.

### 3.1 Theoretical Analysis

In this section, we derive an upper-bound on the excess risk $\mathcal{R}_n(\mathcal{A}_{\mathrm{opt}})$ of Algorithm 2. We also show that the performance achieved this algorithm cannot in general be improved, by obtaining an algorithm independent lower-bound on the excess risk for a particular class of problems.

**Upper Bound.** We begin by obtaining an upper bound on the convergence rate of the excess risk of $\mathcal{A}_{\mathrm{opt}}$, the proof of this result is in Appendix B.1.

**Theorem 1.** *Let Assumptions 1-3 hold and define* $\pi_{\min} := \min_{z \in \mathcal{Z}} \pi^*(z)$. *Fix any A such that* $\pi_{\min}/2 \leq A < \pi_{\min}$. *Suppose the query budget* $n$ *is sufficiently large, as defined in Equation 19 in Appendix B.1. Then, with probability* $1 - \delta$, *the excess risk of Algorithm 2 can be upper-bounded as*

$$\mathcal{R}_n(\mathcal{A}_{opt}) = \max_{z \in \mathcal{Z}} L(z, f_{\pi_n}) - M^* = \mathcal{O}\Big( \frac{|\mathcal{Z}|C}{\pi_{\min}} \max_{z \in \mathcal{Z}} e_z(N_A) \Big), \tag{5}$$

*where* $M^* = L(z, f_{\pi^*}), \ \forall z \in \mathcal{Z}$ *(see Proposition 1) and* $N_A = n\pi_{\min}^2/(2\pi_{\min} - A)$.

**Remark 1.** *The uniform deviation bounds,* $e_z(N)$*, defined in Assumption 4 and the bound on the excess risk of Algorithm 2 (see Equation 5) can be instantiated for several commonly used classifiers (function classes* $\mathcal{F}$*) to obtain an explicit convergence rate in* $n$*. If* $\mathcal{F}$ *has a finite VC-dimension,* $d_{VC}$*, then a suitable deviation bound is* $e_z(N) = 2\sqrt{\big(2d_{VC} \log\big(2eN/d_{VC}\big) + 2 \log\big(2N^2\pi^2|\mathcal{Z}|/3\delta\big)\big)/N}$*, where here* $\pi$ *is the numerical constant. And if* $\mathcal{F}$ *has Rademacher complexity* $\mathfrak{R}_n$*, we can choose* $e_z(N) = 2\mathfrak{R}_N + \sqrt{\log\big(N^2\pi^2|\mathcal{Z}|/3\delta\big)/N}$*. Furthermore, with these uniform deviation bounds and if* $\mathfrak{R}_n$ *is* $\mathcal{O}\left(n^{-\alpha}\right)$*, for some* $\alpha > 0$*, then we obtain excess risk bounds of*

$$\mathcal{R}_n(\mathcal{A}_{opt}) = \mathcal{O}\Big( \frac{|\mathcal{Z}|C}{\pi_{\min}^{1.5}} \sqrt{\frac{d_{VC}}{n}} \Big), \qquad \mathcal{R}_n(\mathcal{A}_{opt}) = \mathcal{O}\Big( \frac{|\mathcal{Z}|C}{\pi_{\min}^{1+\alpha} n^{\alpha}} \Big),$$

*for the VC-dimension and Rademacher cases, respectively. These conditions on* $\mathcal{F}$ *are satisfied by several commonly used classifiers, such as linear, SVM, and multi-layer perceptron.*

**Remark 2** (Analysis of $\epsilon$-greedy). *Abernethy et al. (2020) only analyze the greedy version (i.e.,* $\epsilon = 0$*) of their* $\epsilon$-greedy *sampling strategy with* $|\mathcal{Z}| = 2$*, and show that at time* $n$*, either the excess risk is of* $\mathcal{O}\big( \max_{z \in \mathcal{Z}} \sqrt{2d_{VC}(\ell \circ \mathcal{F}) \log(2/\delta)/N_{z,n}} \big)$*, or the algorithm draws a sample from the attribute with the largest loss. However, due to the greedy nature of the algorithm they analyzed, there are no guarantees that* $N_{z,n} = \Omega(n)$*, and thus, in the worst case the above excess risk bound is* $\mathcal{O}(1)$*. We show in Appendix C how the techniques we developed for the analysis of Algorithm 2 can be suitably employed to study their* $\epsilon$-greedy *strategy. In particular, we obtain sufficient conditions on* $\epsilon$ *under which the excess risk of* $\epsilon$-greedy *converges to zero, and the rate at which this convergence occurs.*

**Remark 3** (Comparison between $\mathcal{A}_{\mathrm{opt}}$ and $\epsilon$-greedy). *$\mathcal{A}_{opt}$ holds two primary advantages over* $\epsilon$-greedy*, these are discussed in detail in Appendix C, but briefly* $\mathcal{A}_{opt}$ *has superior risk guarantees, by a constant factor. And both algorithms depend on tunable parameters:* $C$ *for* $\mathcal{A}_{opt}$ *and* $\epsilon$ *for* $\epsilon$-greedy*, but* $\mathcal{A}_{opt}$ *is far more robust to the choice of parameter. Figure 9 in Appendix C illustrates a case where poor choice of* $\epsilon$ *can prevent* $\epsilon$-greedy *from converging to the optimal mixture distribution for any number of samples. This robustness makes* $\mathcal{A}_{opt}$ *easier to employ in practical settings.*

**Lower Bound.** Let $\mathcal{Q} = (\boldsymbol{\mu}, \mathcal{F}, \ell_{01})$ denote the class of problems where $\boldsymbol{\mu} \in \mathcal{M}$ is an instance of the `SyntheticModelI` described in Section 2.2, $\mathcal{F}$ is the class of linear classifiers in two dimensions, and $\ell_{01}$ is the $0-1$ loss. For this function class $\mathcal{F}$, $e_z(N) = \mathcal{O}(\sqrt{\log(N)/N})$ for $z \in \mathcal{Z} = \{u, v\}$, which implies that the excess risk achieved by both $\mathcal{A}_{\text{opt}}$ and $\epsilon$-`greedy` strategies is of $\mathcal{O}\left(\sqrt{\log(n)/n}\right)$. We prove in Appendix D that this convergence rate (in terms of $n$) cannot in general be improved by showing that $\max_{Q \in \mathcal{Q}} \mathbb{E}_Q [\mathcal{R}_n (\mathcal{A})] = \Omega (1/\sqrt{n})$.

## 4 Heuristic Extensions

We show in Appendix B.1.1 that if the uniform deviation bounds, $e_z(N)$, can be chosen to decrease to zero sufficiently quickly as $N$ increases, then we can omit the $C$ dependent third term in Equation (4) and still attain the same regret bounds given in Theorem 1. The resulting two-term UCB is then only the high probability upper bound on $L(z, \hat{f}_t)$. We will use this UCB as the basis for several practical modifications to our optimistic adaptive sampling strategy. First we note that in UCB based algorithms the confidence bounds necessary to attain theoretical results often do not produce optimal empirical results–see, for example, Section 6 in Srinivas et al. (2009). So, following standard practice, we introduce a hyperparameter, $c_0$, in Equation (6) below, which we can tune to optimize the exploration/exploitation trade-off.

Practical applications of our strategy run into two further challenges. As described in Section 3, $\mathcal{A}_{\text{opt}}$ requires re-training the classifier in every iteration. While this can be implemented in small problems, it becomes infeasible for problems involving large models, such as CNNs. Also, as noted in Section 2.1, for data where one attribute is significantly easier than the other, Assumption 2 may not hold. In this case it may not be beneficial to continue to sample from the attribute with the largest loss. To address these issues we present a heuristic variant of Algorithm 2.

For the first challenge we make the following modifications to Algorithm 2. **1)** We expand Phase 1 to $n_0$ rounds, **2)** at each subsequent round we draw two batches of size $b_0$ from the chosen attribute, **3)** instead of re-training from scratch each round, we update the previous model, $\hat{f}_{t-1}$, and **4)** instead of training to convergence, we perform one gradient step over the entire accumulated training set, $\mathcal{D}_t$.

To address the second challenge we modify the UCB by adding a term based on the Mann-Kendall statistic (Mann, 1945) (Kendall, 1948) which, for time series data $(X_i)_{i=1}^n$, is given by $S = \sum_{i=1}^{n-1} \sum_{j=i+1}^{n} \operatorname{sgn}(X_j - X_i)$. This is designed to identify monotonic upwards or downward trends in time series. We calculate the statistic for each attribute, and denote it $S_{z_t}$, for the accuracy of the classifier, $\hat{f}_t$, on the attribute's validation set, $\mathcal{D}_{z_t}$, over time. Intuitively, this is tracking whether training on additional samples from an attribute is improving the classifier's accuracy on that attribute. We incorporate the statistic into the UCB as follows:

$$\widetilde{U}_t(z, \hat{f}_t) \coloneqq \frac{1}{|\mathcal{D}_z|} \sum_{(x,y) \in \mathcal{D}_z} \ell(\hat{f}_t, x, y) + \frac{c_0}{\sqrt{N_{z,t}}} + c_1 \frac{S_{z_t}}{\sqrt{\operatorname{var}(S_{z_t})}}, \tag{6}$$

to incentivize the algorithm to sample attributes on which accuracy is increasing. $c_1$ is a free parameter controlling the importance of per-attribute loss trends. We note that the general ability to modify the UCB in this manner is a strength of our algorithm, as it allows for a great deal of interpretability and adaptability in practical applications.

The parameters $c_0$, $c_1$, $n_0$, and $b_0$ are chosen based on the specific problem and available computational resources. We describe our selection of these terms Appendix E. In the experiments we will refer to the variant of this algorithm with $c_1 = 0$ as $\mathcal{A}_{\text{opt}}$, and the variant with $c_1 > 0$ as $\mathcal{A}_{\text{opt}}+$.

## 5 Empirical Results

We evaluate the performance of our proposed active sampling algorithm $\mathcal{A}_{\text{opt}}$ on both synthetic and real datasets, and compare it with the following baselines: **1)** $\epsilon$-`greedy` scheme of Abernethy et al. (2020), **2)** `Greedy` scheme, which is equivalent to $\epsilon$-`greedy` with $\epsilon = 0$, **3)** `Uniform`, where an equal number of samples are drawn for each attribute, and **4)** `Uncurated`, where samples are drawn according to the natural distribution of the dataset. We note that, for some datasets, the natural distribution is uniform, for those we omit the results for the `Uncurated` scheme.

We will make the code for these experiments available in the supplemental materials. All experiments were run for multiple trials–the number of which is indicated in the results–and we report the average over trials with shaded regions on plots indicating $\pm 1$ standard deviation. Experiments for image datasets were run on a single GPU provided by Google Colab or other shared computing resources.

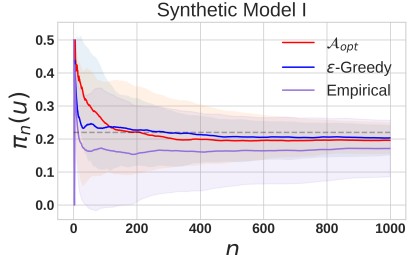
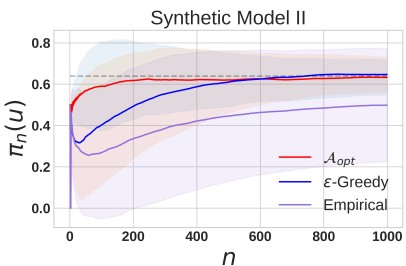

Figure 2: The figure shows the convergence of $\pi_n(u)$ for the three algorithms $\mathcal{A}_{\mathrm{opt}}$, $\epsilon$-greedy (with $\epsilon = 0.1$) and Empirical (i.e., $\epsilon$-greedy with $\epsilon = 0$), to the optimal value $\pi^*(u)$ for the two instances of the SyntheticModel1 introduced in Section 2.2.

**Synthetic Dataset.** In this experiment, we compare how the $\pi_n(u)$ returned by the different algorithms converge to $\pi^*(u)$ for the two synthetic models introduced in Section 2.2 with $\mathcal{F}$ chosen as the family of logistic regression (LR) classifiers. Since the feature space is two dimensional, we use the version of $\mathcal{A}_{\mathrm{opt}}$, $\epsilon$-greedy, and Empirical schemes in which we train (from scratch) the classifier in each round. For $\mathcal{A}_{\mathrm{opt}}$, we use UCB given in 6 with $c_0 = 0.1$ and for $\epsilon$-greedy, we use $\epsilon = 0.1$. These values of $c_0$ and $\epsilon$ are selected via a grid search.

Figure 2 shows how $\pi_n(u)$ changes with $n$ for the three algorithms averaged over 100 trials. As expected, the algorithms with an exploration component (i.e., $\mathcal{A}_{\mathrm{opt}}$ and $\epsilon$-greedy) eventually converge to the optimal $\pi^*(u)$ value in both cases, whereas the Empirical scheme that acts greedily often gets stuck with a wrong mixture distribution, resulting in high variability in its performance.

***Adult* Dataset.** For the remaining experiments we find that, for properly tuned values of $\epsilon$ and $c_0$, both $\mathcal{A}_{\mathrm{opt}}$ and $\epsilon$-greedy attain comparable minimum test error. So we omit the $\epsilon$-greedy results for the purpose of clarity, see Appendix C for a detailed comparison of the two algorithms.

We now analyze the performance of the remaining algorithms on a dataset from the UCI ML Repository (Dua and Graff, 2017) that is commonly used in the fairness literature: the *Adult* dataset. It is a low dimensional problem, so we use LR classifiers and the exact version of each sampling algorithm, where the optimal classifier is computed with each new sample.

We set $\mathcal{Z}$ to be white men, non-white men, white women, and non-white women. The minimum test accuracy over all attributes at each iteration is displayed in Figure 3a. Each sampling scheme approaches $80.5\%$ accuracy as sample size grows, this matches the results achieved under the LR algorithms reported in Table 4a) in Martinez et al. (2020). This is the maximum accuracy an LR classifier can achieve on the white male sub-population in isolation and so additional sampling, or other fairness algorithms, cannot improve on this performance in a minimax sense. We do note, however, that the adaptive algorithms hold a sizable advantage over Uniform at small sample sizes.

**Image Datasets.** We also compare the performance of the different sampling schemes on larger scale problems and more complex hypothesis classes. To this end we use three image datasets: *UTKFace* (Zhang et al., 2017), *FashionMNIST* (Xiao et al., 2017), and *Cifar10* (Krizhevsky and Hinton, 2009) with CNNs. We use the heuristic variants of the $\mathcal{A}_{\mathrm{opt}}$, with $c_1 = 0$, and Greedy algorithms with batch sizes of 50. The CNN architecture, data transforms, and further algorithm parameters are detailed in Appendix E.

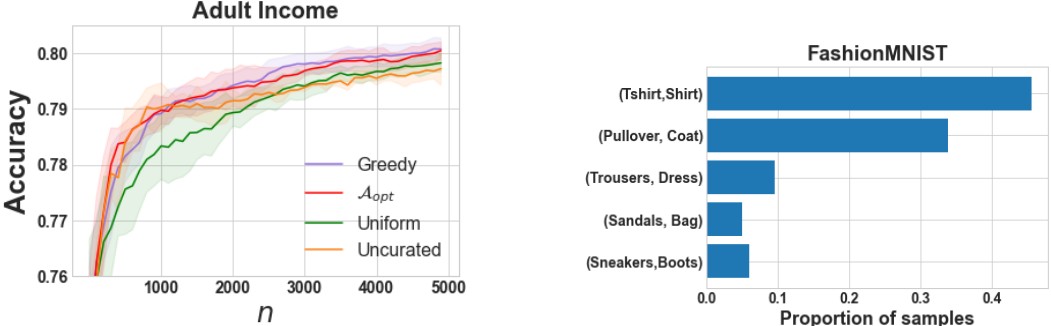

Figure 3: Left: Minimum test accuracy over all attributes for *Adult* dataset as a function of the sampling budget $n$, averaged over 10 trials for *Adult*. Right: Mixture distribution learned by $\mathcal{A}_{\mathrm{opt}}$ over 500 training rounds on *FashionMNIST*.

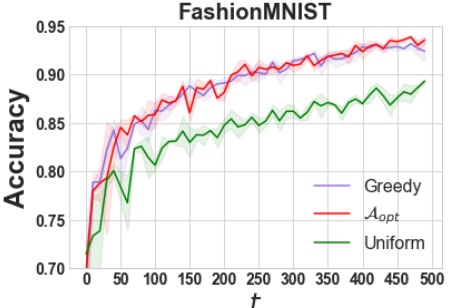
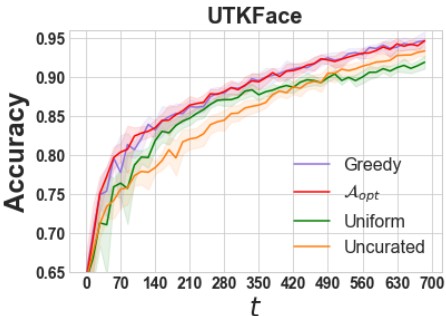

Figure 4: Minimum test accuracy, over all attributes, for both *FashionMNIST* and *UTKFace* as a function of the time step $t$, averaged over 10 trials.

The *UTKFace* dataset consists of face images annotated with age, gender, and ethnicity. We choose $Y = \{\texttt{Male},\texttt{Female}\}$ and set $\mathcal{Z}$ to the five ethnicities. The minimum test accuracy, over all attributes, is shown in Figure 4b and demonstrates a clear separation between the adaptive algorithms and both `Uniform` and `Uncurated` sampling. `Uncurated` does particularly poorly in the low sample regime here, in contrast with the *Adult* dataset, where `Uncurated` performed comparably to the adaptive algorithms. This is because, for *Adult* dataset, the lowest accuracy attribute is over-represented in the dataset with white men at $63\%$ of all samples. Whereas, for *UTKFace*, the accuracy was lowest for Asian people, who are under-represented in the available data at only $14\%$ of all samples.

The other two datasets, *FashionMNIST* and *Cifar10*, were chosen to provide a controlled setting to demonstrate the existence of hard and easy attributes on real-world data. For both, we divide the labels into 5 pairs and assign each an "attribute". The pairs were chosen according to existing confusion matrices to have pairs that are both easy and hard to distinguish from each other, see Appendix E for more details.

For *FashionMNIST*, each pair in Figure 3b is one attribute and within each pair one item is assigned $Y = 0$ and the other $Y = 1$. Then a single binary CNN classifier is learned simultaneously over all 5 pairs. Figure 3b shows the mixture distribution generated by the $\mathcal{A}_{\text{opt}}$ sampling scheme, which allocated the vast majority of samples to the (Tshirt, Shirt) and (Pullover, Coat) pairs. This makes intuitive sense, since both pairs of items are qualitatively very similar to each other, and aligns with common confusion matrices which indicate that those items are frequently misclassified as each other by standard classifiers. Figure 4a displays the worst case accuracy for each scheme on *FashionMNIST* as a function of time step and shows that both adaptive algorithms outperform `Uniform` sampling throughout the training process. Finally, Figure 5 shows the test accuracy on each attribute for both $\mathcal{A}_{\text{opt}}$ and `Uniform` sampling schemes. $\mathcal{A}_{\text{opt}}$ maintains a much smaller spread between the accuracy over different attributes. This equity is particularly desirable from a fairness perspective as it ensures no attribute has a large advantage over any other in the sense of expected accuracy. It also validates our assumption that the optimal distribution will tend to equalize the losses across attributes in real world data.

Final accuracy for all experiments, per-attribute accuracy for *UTKFace*, all results for *CIFAR10*, and details of the *Adult* dataset are included in Appendix E, along with results for the *German* dataset, another UCI ML Repository dataset.

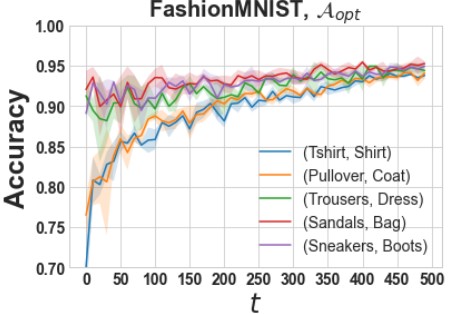
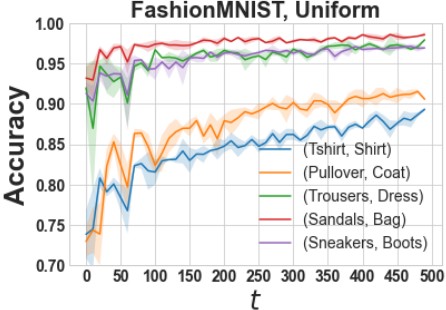

Figure 5: Test accuracy for each attribute in *FashionMNIST* as a function of the time step, $t$, for both $\mathcal{A}_{\text{opt}}$ and `Uniform` sampling schemes, averaged over 10 trials.

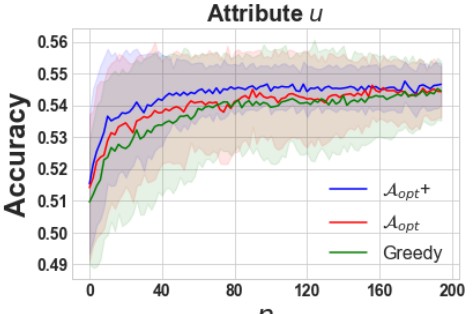
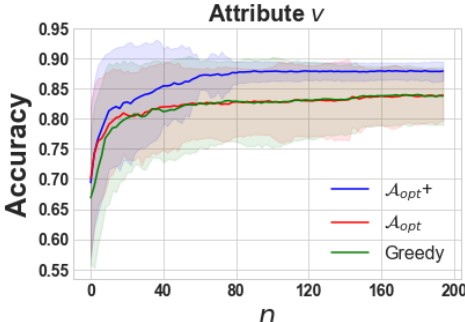

Figure 6: Test accuracy as a function of sampling budget for both attributes from the dataset shown in Fig. 7, averaged over 100 trials.

**Empirical results when Assumption 2 is violated.**

Finally, we consider the case where our assumption of a unique, risk equalizing mixture distribution may not hold. The practical effects of this situation can be seen in the *Adult* dataset where additional samples of the white male attribute show diminishing returns, while other attributes can attain higher accuracy given more samples. In such a scenario, each adaptive algorithm will continue to select samples mainly from the worst case group despite this granting little improvement in performance.

To evaluate this scenario we introduce a second synthetic data model: an instance of `SyntheticModel2` is illustrated in Figure 7 and specified in detail in Definition 4 in Appendix E. In this model we create one attribute, shown in red, which allows a relatively small maximal accuracy that can be attained with few samples. In contrast, the blue attribute can be classified with high accuracy given few sample, as the bulk of its mass is in the central, separable clusters shown in the figure. But classification of this attribute also benefits from many additional samples as the sparser regions in the top left and bottom right are explored. In this setting, constantly sampling from the attribute with the lowest empirical accuracy is inadvisable as additional samples of the red attribute cannot increase its accuracy, while additional samples of the blue attribute can increase its accuracy.

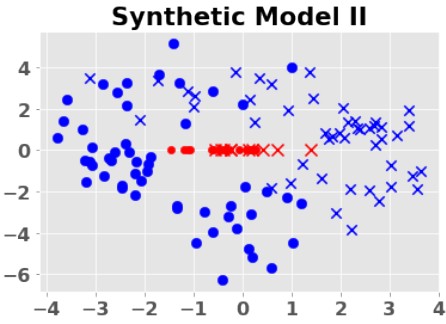

Figure 7: An instance of `SyntheticModel2` with attribute $Z = u$ shown in red and $Z = v$ shown in blue.

We compared the performance of $\mathcal{A}_{opt}+$, the heuristic variation of our algorithm with a trend-based statistic included, $\mathcal{A}_{opt}$, and `Greedy` on `SyntheticModel2`. Figure 6 shows the results, each of the three algorithms achieves similar minimax accuracy and performs worst on attribute $Z = u$, shown in red in Figure 7. But $\mathcal{A}_{opt}+$ achieves 4 points higher average accuracy than both `Greedy` and $\mathcal{A}_{opt}$ with the original UCB on the other attribute, $Z = v$. This demonstrates that the additional term in the $\mathcal{A}_{opt}+$ UCB is effective at recognizing when the hardest group is not benefiting from additional training samples and redistributing them more effectively.

## 6 Conclusion

We considered the problem of actively constructing a training set in order to learn a classifier that achieves minimax fairness in terms of predictive loss. We proposed a new strategy for this problem ($\mathcal{A}_{opt}$) and obtained theoretical guarantees on its performance. We then showed that the theoretical performance achieved by $\mathcal{A}_{opt}$ and $\epsilon$-`greedy` (Abernethy et al., 2020) cannot be improved in general, by obtaining algorithm independent lower-bounds for the problem.

Our experiments demonstrated that adaptive sampling schemes can achieve superior minimax risk [Fig. 3a, Fig. 4] and smaller disparity between per-attribute risks [Fig. 5] compared to both uniform and uncurated schemes. The results in Fig. 2 show the necessity of exploration, as purely greedy algorithms can converge to sub-optimal mixture distributions. And finally Fig. 6 shows the versatility of our general UCB-based strategy in its ability to readily incorporate new terms to accommodate the practical challenges posed by real-world problems.

Our theoretical results rely on the existence of a unique risk equalizing mixture distribution $\pi^*$, a condition that may not always hold. Thus, an important future work is to relax this assumption, and to design and analyze algorithms that identify pareto optimal mixture distributions that achieve minimax fairness. Another important direction is to derive and analyze active sampling strategies for other fairness measures.

