# OpenReview forum: "Adaptive Sampling for Minimax Fair Classification"
_NeurIPS.cc/2021/Conference — NeurIPS 2021 Poster_

### Official Review · Reviewer_iKc6 · 2021-07-11

**Rating:** 8
**Confidence:** 3

**Summary:**

This paper proposes an adaptive sampling scheme to construct the training dataset which allows us to learn classifiers that are fair in a minimax sense. They proved theoretical results on the excess risk of their algorithm, and also proposed heuristic extensions of the algorithm to be applicable to real-world large scale problems. They validate their algorithm on a series of synthetic and real-world datasets, showing an improved performance over empirical/uniform sampling schemes, which also verified their theoretical findings.

**Limitations And Societal Impact:**

No. This work is mostly theoretical work on how to design a sampling scheme to drive fair classification. I do not think it is related to any negative societal impact.

**Main Review:**

Overall this is a very well-written paper. I especially like the results that the minimax optimal mixture distribution yields a classifier that has equal risk among all attributes. The theoretical analysis is convincing, showing the quality of the solution from Algorithm 1 via the excess risk measure. The supporting examples well illustrate the validity of the assumptions. They also give practical suggestions on what to do when certain assumption is violated, and how to properly modify the algorithm to be suitable for real-world large-scale problems.

I have a question on Theorem 1, where eq. (5) has this term e_z(N_A). Since A could be any value between \pi_{min}/2 and \pi_{min}, can we choose an optimal A such that the term e_z(N_A) will be the smallest for all z? Also in practical implementations, what is the form of e_z(N) that is used in the UCB (4)?

In Eq. (6), is S_{z_t} calculated as the Mann-Kendall statistic of the accuracy of f_t? That is, replacing the terms X_i, X_j in the expression of S by Acc(f_t)? Does the c_0 term of Eq. (6) account for the uniform deviation term e_z of Eq. (4)? Having var(S_{z_t}) in the denominator suggests that we should draw more samples for attributes whose var(S_{z_t})  is small. But if the accuracy for a certain attribute is already very good (cannot be further improved by drawing more samples), in which case var(S) would be small, drawing more samples for such attributes seems to be counterintuitive?

My major concern in the experimental section is that the performance of \epsilon-greedy seems to be very close to A_{opt}. In Fig. 2 (left), \epsilon-greedy converges even faster (much faster). In Figs. 3 and 4 the greedy algorithm also has a similar performance to A_{opt}. Do these results suggest that \epsilon-greedy and A_{opt} have comparable performance in reality? Can you provide guidance on which one to use?

**Time Spent Reviewing:**

6

---

> ### Author Response · Authors · 2021-08-10
> **Response to reviewer iKc6**
>
> We thank the reviewer for their thoughtful evaluation of our work and appreciate their questions and comments.  We will attempt to answer all the questions below, and we are happy to follow-up in the discussion period.
>
> #### **Details of $e_z$**
> For Theorem 1, we can choose $A$ to obtain tighter deviation bounds, in general we would want to choose $A$ as small as possible since $e_z$ must be non-increasing, and so will be smaller the larger $N_A=n\pi_{min}^2(2\pi_{min}-A)$ is.
>
> For practical implementations of our algorithm, for some problem settings we can explicitly calculate what the $e_z$ are, as given in Line 226 for the case of a function class with finite and known VC-dimension.  But in general we implement the algorithm by adopting the tunable $c_0$ parameter in Equation 6.  We will add a more thorough discussion of this in the final manuscript, but in brief, it is common in UCB-based algorithms that the form of the confidence bound necessary for theoretical results does not give optimal performance in practice (for instance, see Sec. 6 of Srinivas et al. (2009) regarding the experimental protocol of the popular GP-UCB algorithm for Bayesian Optimization). So, we followed standard practice and maintained the $n^{-1/2}$ dependence while introducing and optimizing $c_0$ to accomodate the rest of the bound and allow for better performance.
>
> #### **Mann-Kendall statistic**
> For the Mann-Kendall statistic, we are tracking the accuracy for each attribute separately, so we have $|\mathcal{Z}|$ separate statistics, where $S_{z_t}$ tracks the accuracy of attribute $z$. Thus, $X_i$ should be replaced with the accuracy of the classifier $f_t$ only on samples from attribute $z$. But the important point here is that the statistic is based on *trends* in the accuracy, so neither $S_{z_t}$ or its variance is characterizing the accuracy itself -- instead it is characterizing how the accuracy is responding to additional samples from $z$. This tells us that if $S_{z_t}$ is large and positive and its variance is small, then we're very confident that the accuracy on attribute $z$ is increasing and so it's productive to continue to draw samples from that attribute.
>
> But if the statistic is small or negative or has high variance, additional samples of the attribute are not giving a significant, consistent increase in accuracy. This is the scenario that motivated the development of $\mathcal{A_{opt}}+$. It could occur that the hardest attribute, with the worst accuracy, simply cannot be improved upon by seeing more samples -- as in the case of the red attribute in Figure 1c. $\mathcal{A_{opt}}$ and $\epsilon$-greedy, would continue to fruitlessly sample from this hard attribute. But in $\mathcal{A_{opt}}+$, the Mann-Kendall statistic recognizes that the loss has stagnated and so decreases the associated UCB, allowing other attributes to be sampled. We will make this more clear in the final version of the paper.
>
> #### **Comparison of $\epsilon$-greedy and $\mathcal{A}_{opt}$**
> In the case where the parameters of both algorithms have been well-tuned, they generally do attain similar performance -- for the experiments that included both we ran grid searches to find the optimal values of $\epsilon$ and $c_0$. However, as shown in Figure 9 in the appendix, if $\epsilon$ is chosen poorly the $\epsilon$-greedy algorithm may not converge to the optimal mixture distribution at all. As discussed in Appendix C.4, $\mathcal{A}_{opt}$ is much more robust to the value of its parameter and so is easier to use.
>
> $\mathcal{A_{opt}}$ also has a practical advantage in that its exploration is intelligent, while $\epsilon$-greedy explores randomly.  So, for instance, in the case where classification of two attributes is much more difficult than any of the other attributes, $\mathcal{A_{opt}}$ will almost always sample from the two difficult attributes, while $\epsilon$-greedy would be forced to also sample from the easy attributes, which is less efficient in a sense.  We chose not to include mention of this in the paper as we could not find a dataset that demonstrates this behavior well -- but in such a scenario $\mathcal{A_{opt}}$ should outperform $\epsilon$-greedy, even given an optimal value for $\epsilon$.
>
>
> #### References:
> * Srinivas et al. (2009). _Gaussian Process Optimization in the Bandit Setting: No Regret and Experimental Design_. ICML 2010.

---

### Official Review · Reviewer_Br95 · 2021-07-12

**Rating:** 6
**Confidence:** 3

**Summary:**

The paper exploits active sampling methods to construct a training dataset that yields fair classifiers in a minimax sense: minimising the maximum risk of any of the groups of interest. The proposed method constructs a dataset iteratively by drawing samples from the group with highest upper confidence bound (UCB) for the predictive risk of the current classifier. At each iteration, the classifier is retrained (updated) on the extended dataset, and the process is repeated until a sampling budget is exhausted. The authors provide theoretical bounds for the convergence of the algorithm, which are proven to be tighter than those of previous $\epsilon$-greedy approaches.

**Limitations And Societal Impact:**

The authors did not address any potential negative societal impact in their paper. However, given that their methods strive to achieve fairness, I would imagine the societal impact would be mostly positive. The authors do address some of the limitations of their approach, and I do not have anything to add to their considerations.

**Main Review:**

[Overall]
- *Originality*: The proposed methods and theoretical results are, to the best of my knowledge, novel.
- *Quality*: The paper seems technically sound, and the theoretical analysis is thorough with interesting results. The experimental results, however, are less encouraging.
- *Clarity*: While the paper is well written, the main contributions are not made immediately obvious. Most of the promising results are hidden in the appendix.
- *Significance*: Fairness is a topic of great interest in the machine learning community, and I believe the proposed active sampling approach is a promising avenue to tackle it.

[Strengths]
- I find the idea of combining minimax Pareto fairness with active sampling quite promising.
- The theoretical analysis is thorough and the convergence results are a valuable contribution to the community.
- I appreciate that the authors provided their code in a fairly accessible format.

[Weaknesses]
- The experimental results seem weak. The proposed method does not seem to outperform the previous simpler $\epsilon$-greedy approach and the other baselines, uncurated and uniform, are rather weak.
- The text is well written but not particularly easy to follow. Given the interesting theoretical results, I think the paper would read much better if the focus was put in the theoretical development—with at least sketches of the proofs in the main text—rather than on the experimental results which are not as promising or revealing.

[Further Comments and Questions]
- In Figure 7, the performance of $\mathcal A^+_{opt}$ seems quite promising. Why is it not included in the other experiments?
- I think Assumption 3 should also be stated on the main text.
- I believe there is a typo in line 232 that probably should read '[...] the performance achieved *by* this algorithm cannot in general be improved [...]'.

**Time Spent Reviewing:**

4

---

> ### Author Response · Authors · 2021-08-10
> **Response to reviewer Br95**
>
> We appreciate the reviewer's thorough evaluation of both the strengths and weaknesses of our work as well as their helpful questions and suggestions.  We will attempt to answer their questions below, and would be happy to follow-up in the discussion period.
>
> #### **Choice of baselines**
> Our work is focused on the data collection/dataset construction portion of the ML pipeline, which is a relatively new approach to fairness.  Most other fairness algorithms try to impose constraints during the training phase or run some kind of post-processing to fix biased outputs.  So these approaches aren't directly comparable to ours, and in fact could be used in conjunction with ours -- studying the composition of fairness methods applied to different portions of the ML pipeline is an interesting direction for future work.  But, to the best of our knowledge, we've included all directly comparable works in our paper.
>
> Beyond that, we compared against the Uncurated, Uniform, and Greedy schemes as they are intuitive, if simple, approaches to the problem and they provide verification that there is indeed value in applying the more complicated $\epsilon$-greedy and $\mathcal{A_{opt}}$ algorithms during the data collection phase.  The Uniform and Greedy schemes are not necessarily weak--if bias were solely the result of imbalanced training data then the Uniform scheme would solve the problem, or if it were sufficient to always sample from the attribute with the lowest accuracy, the Greedy scheme would be optimal.  So the experiments with these baselines establish the necessity of an adaptive, explorative algorithm like $\epsilon$-greedy or $\mathcal{A_{opt}}$.
>
> #### **Comparison of $\epsilon$-greedy and $\mathcal{A}_{opt}$**
> We first note that we considerably expanded on the theoretical analysis of the $\epsilon$-greedy algorithm, as well as developed our own $\mathcal{A_{opt}}$ algorithm.  So our goal here was not solely to demonstrate the superiority of $\mathcal{A_{opt}}$. That being said, $\mathcal{A_{opt}}$ does have theoretical advantages over $\epsilon$-greedy, as described in Remark 1, as well as tighter, by a constant factor, regret bound.
>
> Empirically, when their parameters are well-tuned, the two algorithms do have similar performance.  However, as shown in Figure 9 and discussed in Appendix C.4, a poor choice of $\epsilon$ can prevent the $\epsilon$-greedy algorithm from finding the optimal mixture distribution. In general $\mathcal{A}_{opt}$ is much more robust to parameter selection, which makes it easier to implement.
>
>
> #### **$\mathcal{A}_{opt}+$**
> $\mathcal{A_{opt}}+$ was not included in the majority of our experiments as it lacks the theoretical grounding that $\mathcal{A_{opt}}$ has and the real world datasets that we worked with generally did not require it. As shown in Figures 6, 10, and 12, it appears that the assumption that the optimal mixture distribution yields equal accuracies across all attributes is realized in these datasets.  But we appreciate the reviewer's comment and agree that $\mathcal{A_{opt}}+$ merits additional study, both theoretically and empirically, in future work.
>
> #### **Theoretical development**
> We agree that Assumption 3 should be stated in the main text of the paper, along with more of the theoretical development.  We intend to use the extra page available for the final manuscript to restructure the presentation of our theoretical arguments and include a more thorough discussion in the main text.

---

> > ### Comment · Reviewer_Br95 · 2021-08-22
> > **To the authors' comments**
> >
> > I thank the authors for responding to my questions and comments. I am satisfied with most of their answers, but I just would like to reiterate my opinion that the paper could be better organised. A more thorough discussion of the theoretical results in the main text will be beneficial, but the experiments in appendix C.4 could also be mentioned in the main text. The point that $\mathcal A_{opt}$ avoids the issue of fine-tuning $\epsilon$ could be highlighted, as it showcases an empirical advantage of $\mathcal A_{opt}$ that is not evident in the main text.

---

> > > ### Author Response · Authors · 2021-08-24
> > > **Response to reviewer Br95**
> > >
> > > We appreciate the reviewer reading our response and are glad we have been able to provide satisfactory answers to their questions.  We agree with their general comments about the organization of the paper and will include more explicit, expansive discussion of the advantages $\mathcal{A}_{opt}$ has over $\epsilon$-greedy, including the sensitivity to parameter tuning, in the main paper for the final manuscript.

---

### Official Review · Reviewer_LCx4 · 2021-07-15

**Rating:** 6
**Confidence:** 2

**Summary:**

This paper proposes a sampling algorithm to construct the training dataset adaptively, the goal is to learn a classifier that can attain minimax fairness measured by the predictive loss. The authors show that the proposed algorithm has nice theoretical properties (upper bound on the regret, and this upper bound can not be improved).

**Limitations And Societal Impact:**

This paper is on fairness, so I encourage the authors to discuss some risk, limitations and promise of the A_opt sampling algorithm.

**Main Review:**

I enjoyed reading the introduction and the problem formulation part of this paper. Nevertheless, I encounter difficulty to digest section 3 which contains the main contribution of this paper:

- The paper lacks some intuition on the sampling algorithm. Can the author provide some intuitive explanation on why Algorithm 1 may work?
- In line 226, why does the expression involve n but not N?
- What is the query budget n so that Theorem 1 is valid (say, for Adult and German dataset and a logistic/SVM classifier)?
- Empirically, it is difficult to be convinced that A_opt is a better algorithm than the eps-greedy method of Abernerthy et al. (2020). Apart from the theoretical difference in Remark 1, can the authors provide any justification to use A_opt?


Minor comments:
- In line 199, xi is in (0, 1) but in line 218, xi is in (0.5, 1)?


**Time Spent Reviewing:**

6

---

> ### Author Response · Authors · 2021-08-10
> **Response to reviewer LCx4**
>
> We would like to thank the reviewer for their evaluation of our paper and their questions and suggestions. We will try to respond to the questions below and are happy to follow up during the discussion period.
>
> #### **Intuition for Algorithm 1**
> We plan to use the additional page available for the final manuscript to bring more theory and exposition into the main paper, which should make the intuition behind the algorithm more clear. But, in brief, the intuition is that, at any given time step, the UCB for each attribute gives an indication of how large the loss of the current classifier could plausibly be on the associated attribute.  This is because the first term of the UCB is an empirical estimate of the loss and the second term is constructed to be a high probability upper bound on the deviation between the true and empirical loss. As such, our algorithm draws samples from the attribute with the largest UCB.  This serves two purposes: more samples of that attribute should increase the accuracy of the classifiers learned in the subsequent rounds *on that attribute* and more samples in the associated validation set, the $D_z$, improves our estimate of the loss on that attribute.  So by executing this process we ensure that we get a sufficiently accurate estimate of the loss on each separate attribute and these estimates allow us to sample more, and so learn better, the attribute with the highest loss.
>
> #### **Comparison of $\epsilon$-greedy and $\mathcal{A}_{opt}$**
> We note that we significantly extended the theoretical analysis of the $\epsilon$-greedy algorithm to establish regret bounds under much less restrictive assumptions, so our goal was not solely to establish a superior algorithm. That being said, in addition to the advantage mentioned in Remark 1, $\mathcal{A_{opt}}$ has tighter, by a constant factor, regret bounds than $\epsilon$-greedy.
>
> Empirically, these two algorithms do have comparable performance in the special case that their respective parameters have been well-tuned: for all experiments presented in the paper we chose $\epsilon$ and $c_0$ by grid search.  But, as shown in Figure 9 in the appendix, if $\epsilon$ is not appropriately tuned, the $\epsilon$-greedy strategy may be unable to converge to the optimal solution at all, while $\mathcal{A_{opt}}$ is much more robust in its dependence on the parameter tuning and so is easier to use.
>
> $\mathcal{A_{opt}}$ also has a practical advantage in that its exploration is intelligent, while $\epsilon$-greedy explores randomly.  So, for instance, in the case where classification of two attributes is much more difficult than any of the other attributes, $\mathcal{A}_{opt}$ will almost always sample from the two difficult attributes, while $\epsilon$-greedy would be forced to also sample from the easy attributes, which is less efficient in a sense.  We chose not to include mention of this in the paper as we could not find a dataset that demonstrated this behavior well.
>
>
> #### **N dependence in line 226**
> This was a typo and we thank the reviewer for catching it, it should be a function of $N$, not of $n$.
>
> #### **Allowed values of $\xi$**
> The instance in Line 218 is a typo: it should state that $\xi$ may be in $(0,1)$.

---

### Official Review · Reviewer_3tnc · 2021-07-16

**Rating:** 4
**Confidence:** 4

**Summary:**

This paper considered solving the minimax fair classification by adaptively constructing a training sets, based on the principle of optimism in the face of uncertainty. The authors provide a simple and straightforward analysis on the performance of ERM on constructed training sets, and empirically show the performance on synthetic and real dataset.


**Limitations And Societal Impact:**

The authors have properly addressed them.

**Main Review:**

Overall the main idea of the paper is clear to me. However, I have several questions on the algorithm design, theoretical analysis.

General Questions:

1. The algorithm assume we have an access to the oracle $P_z$, but in practice we only have an access to a batch of samples, which cannot guarantee we can always get independent samples from $P_z$ (for example, we requires the amount of samples from $z_i$ larger than the total amount of samples that have attribute $z_i$ from the dataset), how can we deal with such scenario?

2. What’s $\pi$ in the definition of $e_z$ (Line 226)? Is it related to $\pi_z$? I don’t see any of clarification throughout the whole Sec 3.

3. It should be better mentioned that $\mathcal{D}_{z}$ is used for cross-validation in Sec 3 and algorithm box.

4. I would like to ask how general is the Assumption 3. The derivation heavily depends on this Assumption and I don’t feel it quite natural. It assumes a reverse of the triangle inequality, but can there be cases that each component is large but have different signs so the sum is small? Can such assumption be replaced with local smoothness and convexity of $f$ and $l$? Such assumption seems more general to me.

5. I would gonna argue that the lower bound is not “matching”, especially the upper bound is linear in $|\mathcal{Z}|$. The lower bound in $n$ is not surprising, as for generalization an $\Omega(n^{-1/2})$ minimax error is unavoidable. I expect to know whether this $|\mathcal{Z}|$ is unavoidable, or is just an artifact introduced by the proof strategy.

Some questions regarding the proof and proof organization.

6. In Line 602, the authors say that the right inequality is due to Lemma 1, but here  $z$ is in the underrepresented group, while Lemma 1 discuss the $z$ in the overrepresented group (see Line 569 and 592), can the authors make a clarification on that, as it’s critical to the proof?

7. I suggest the authors refine the proof in the Appendix. Its’ pretty hard to check when the proof is not well-organized. For example, in Line 603-610, I don’t quite see why by definition $t_0 \geq \pi_{\min} n$ until I see $t_0\geq N_{z_0, t_0}\geq \pi_{\min} n$ and aware of the final inequality is due to forced exploration. In fact, the authors does not emphasize the need of forced exploration in any place.

8. Also, the introduction of terms in (11) and (12) is not so intuitive and easy to follow. It’s not clear where and why should we introduce $A$ (and $A_i$ in the subsequent part), and why we need $n_0 > \pi_{\min}^{-2}$. It’s better to give more explanation to the potential readers.

9. In Line 640 why $\mathcal{Z}_{u}$ could be empty? Please make the claims as accurate as possible.

10. Is $N$ in Line 687 $N_A$ and no $A$ in the right side?

Minor Questions on Experiments.

11. I feel there are some gaps between theory and experiments. It’s understandable to select some confidence parameters when conducting neural nets experiments as the function class complexity is generally not so applicable, but for me it does not support the theoretical results any more.

12. I would also like to ask when we don’t have the oracle $P_z$,what can we do instead. This is the real scenarios we face in practice. As the authors conduct some experiments with real world dataset, I want to mention that the target for such scenarios is totally different from the problems the authors have described.

Overall, this paper is an adaption of the popular UCB method to the minimax fair classification tasks. The main technical difficulty is that, adding new samples from one attribute will affect the performance on all of the attributes, which is different from the bandit setting. The authors address this issue with several new assumptions to prove the optimism and back to the bandit style proof. Assumption 1 is sound, however, assumption 2 is quite restricted and the authors only provide a heuristic fix without any further theoretical justification. Assumption 3 is quite strong without intuitive clarification. The proof is not well-organized and may have some potential issues. Moreover, the algorithm is somehow idealized in the sense that it assumes the data generating process oracle, which is probably not a good setting in practice. Based on the reasons I mentioned above, I tend to reject this paper.


**Time Spent Reviewing:**

3

---

> ### Author Response · Authors · 2021-08-10
> **Response to reviewer 3tnc**
>
> We appreciate the very detailed, thorough review and specific questions and suggestions offered by the reviewer. Below we respond to the questions raised and expand on several opaque points in the paper, we will include these elaborations in the final manuscript. We hope the reviewer finds our response satisfactory and we are happy to follow-up during the discussion period.
>
> #### **Oracle (Comments #1 and #12)**
> The reviewer describes the scenario of _passive supervised_ learning where the learner is provided with a labelled training set that must be used for learning the classifier. In our paper, we consider a scenario where the learner has some control over the training-set construction process. This control is modeled through the access to an oracle that returns a pair $(X,Y) \sim P_{XY|Z}(X, Y|Z=z)$ corresponding to some protected attribute $z$. In practice, this oracle can be simulated either through some data-collection agents (for example a polling agency sending its agents to collect samples from specific protected groups) or via access to a large database of samples labelled by the desired attribute.
>
> In the case that we are drawing from a database that runs out of samples of a given attribute, we take this as indication that the existing data is insufficient for this notion of fairness and particularly that there is a dearth of examples of that attribute. In this event we could then attempt to acquire new samples in an efficient manner -- since we know which attribute we need more samples of.
>
> If we lack access to the oracle $P_z$ entirely while curating the training dataset, it seems there is little that can be done to ensure fairness by dataset construction.  Fortunately, in many datasets, including the Adult and German datasets used in the paper, many pieces of demographic information that are commonly considered protected attributes are available in the feature data.  Please note, we only require access to this oracle at training time and do not need access to the protected attribute when performing classification, which avoids the potential ethical concerns with the use of protected attributes to perform classification.
>
> #### **Definition of $e_z$ (Comment #2)**
> The $\pi$ in the definition of $e_z$ is just the numerical constant, $\pi=3.14...$,  we will add a comment clarifying the conflict in notation.
>
> #### **Cross-validation set (Comment #3)**
> We thank the reviewer for noting that line 194 doesn't make explicit mention of *cross-validation*, we'll add this language to the final manuscript.
>
> #### **Assumption 3 (Comment #4)**
> We thank the reviewer for their suggestion to reframe this assumption in terms of convexity and continuity properties. In fact, we can show that Assumption 3 can be guaranteed if the following two conditions hold:
> * For a fixed $\pi$, the mapping $f \mapsto\mathbb{E}\_{\pi} [ \ell (f, X, Y)]$ is strictly convex, and
> * There exist $C, \epsilon_0>0$ such that for any $f \in \mathcal{F}$ with $\|f - f_\pi\| \leq \epsilon_0$, we have $\mathbb{E}\_{\pi}[\ell(f, X, Y)] - \mathbb{E}\_{\pi}[\ell(f_{\pi}, X, Y)] \geq C \|f - f_{\pi}\|$, where $\|\cdot \|$ is some norm on the function space $\mathcal{F}$.
>
> The first condition ensures the existence of  a unique $f_{\pi}$ for a given value of $\pi$, while the second condition ensures that if a function $f$ is close to $f_{\pi}$ in terms of the expected loss, it must also be close in (some appropriate) norm on the function space $\mathcal{F}$. This closeness in norm along with the continuity from Assumption 1 implies the result. We can check that these conditions are satisfied by linear classifiers with convex loss functions and continuous distributions $P_{X|YZ}$, and can be generalized to possibly infinite dimensional domains through the standard _kernel trick_.
>
> We also believe that Assumption 3 (or some variant) is necessary for the problem considered in our paper to be theoretically tractable. The reason for this is that in our set-up, we learn a single classifier (for all attributes $z$) by empirical risk minimization (ERM), and we wish to ensure minimaxity of the losses incurred by such a classifier over the individual protected attributes. We believe that it may be difficult to obtain such guarantees in situations that allow two classifiers to have near-optimal ERM loss, but very different individual components. We will include a detailed discussion on this in the final manuscript.
>
> #### **Lower bound dependence on $|\mathcal{Z}|$ (Comment #5)**
> We thank the reviewer for noting this. The upper and lower bounds do match if the lower-bound is constructed in full generality. For continuity with our synthetic example, we constructed a lower-bound in the case of two protected attributes (i.e. $|\mathcal{Z}|=2$). The same argument with $m=|\mathcal{Z}|>2$ carries over for a class of synthetic models consisting of $m$ protected attributes and $m$ dimensional Gaussian distributions $P_{X|YZ}$.  And given this construction we attain a lower-bound that also has a linear dependence on $|\mathcal{Z}|$, matching the upper-bound and showing that this dependence is unavoidable. We will adjust the paper to present the lower-bound in full generality and make this dependence clear.
>
> #### **Proof details (Comments #6 and #9)**
> We agree that this was not sufficiently well explained and thank the reviewer for noting this, we will clarify the explanation in the final version. Regarding the claim in Line 602, note that Lemma 1 also tells us something about points in $\mathcal{Z_u}$-- the fact that at time $t_0$ a point $z_{t_0} \in \mathcal{Z_o}$ was selected means that the UCB index $U_{t_0}(\hat{f_{t_0}}, z)$ for any $z \in \mathcal{Z_u}$ must be no larger than the index $U_{t_0}( \hat{f_{t_0}}, z_{t_0})$. Thus the right inequality above Line 602 is a result of the following chain: $$ L(z, f_{\pi_{t_0}}) \stackrel{(i)}{\leq} U_{t_0}(\hat{f_{t_0}}, z) ] \stackrel{(ii)}{\leq} U_{t_0}(\hat{f_{t_0}}, z_{t_0}) \stackrel{(iii)}{\leq} M^* + B,$$ where $(i)$ follows from the UCB definition, $(ii)$ is due to the fact that the point $z_{t_0} \in \mathcal{Z}_o$ was selected by the algorithm at time $t_0$ and $(iii)$ is due to Lemma 1.
>
>
> For the mention of $\mathcal{Z}_u$ in Line 640, $\mathcal{Z}_u$ could be empty there if and only if the learned mixture distribution is equal to the optimal distribution: if and only if $\hat{\pi}_n = \pi^*$.
>
> #### **Proof organization (Comments #7 and #8)**
> We appreciate the reviewers suggestions regarding the organization and clarity of our proofs.  We will re-organize the proofs and add additional context and explanation, particularly around the points they mentioned.
>
> #### **Line 687 typo (Comment #10)**
> The reviewer is correct, Line 687 should read $N_A = \big(\frac{\pi_{min}}{2\pi_{min}-A}\big)n$.

---

### Author Response · Authors · 2021-08-24
**Authors-Reviewers Discussion**

Dear Reviewers,

We believe that we properly and factually addressed all major points raised in your reviews. Please let us know if you have any other concerns. We would be happy to discuss them.


Sincerely,
The authors

---

### Decision · Program_Chairs · 2021-09-27

**Decision:**

Accept (Poster)

**Comment:**

While the review scores are a bit divergent (one reviewer scored 4 while others gave greater than or equal to 6), the reviews are overall positive, and some of the major concerns regarding the organization of theoretical guarantee and the advantage of ${\cal A}_{opt}$ are properly addressed during the discussion period.

Re. the access to the oracle: I believe this assumption is not very strong in light of practical scenarios, although it would definitely be better if the algorithm does not depend on it. Also, the authors explained in the rebuttal how to react to such challenging scenarios properly.

Overall I believe this paper is worth being published, given that the theoretical result is further elaborated in the main text as well as the advantage of ${\cal A}_{opt}$ is highlighted with sufficient experimental supports, as the authors promised.